# BRIM: BLOCK-WISE RETURN INDUCTION METHOD FOR SEQUENCE KNOWLEDGE DISTILLATION

## ABSTRACT

Reinforcement Learning (RL)-based knowledge distillation (KD) is increasingly used to train language models for text generation. However, existing methods suffer from high variance caused by long action chains during sampling. To address this, we propose a novel block-wise return induction approach (called BRIM) that mitigates the high variance issue and stabilizes the training process. Our idea is to apply the Bellman Optimality Equation inversely to each $K$-step block segmented student's explored trajectories, and thus induce a total reward for all blocks from the teacher model, serving as the policy-gradient training signal. Theoretical analysis shows that our BRIM reduces the variance of the gradient estimates, thus leading to improved RL optimization, especially when the student model size is large. Empirical evaluation on three text generation tasks demonstrates that our approach yields superior performance in both standard task metrics and large language model (LLM)-based evaluation, which suggests that our BRIM offers a promising direction for enhancing RL-based KD in LLM research. [1]

## 1 INTRODUCTION

Knowledge distillation (KD; Hinton et al., 2015) refers to training a (typically) small student model from a teacher's output. KD has been increasingly important in the LLM era, as larger models achieve higher performance (Kaplan et al., 2020) but are more difficult to deploy in low-resource scenarios.

KD approaches can be generally categorized into two types: intermediate-layer matching and prediction matching. Intermediate-layer matching aims to match the student's and teacher's hidden states, encouraging the student to mimic the teacher's behavior layer by layer (Sun et al., 2019; Jiao et al., 2020; Wang et al., 2021). Prediction matching informs the student of the task to solve, typically by minimizing the divergence of output distributions (Kim & Rush, 2016; Wen et al., 2023).

Classic KD for text generation suffers from the exposure bias problem (Bengio et al., 2015), as the student learns word by word following the teacher's or ground truth's prefix, without accounting for its own previous predictions. RL alleviates this issue by enabling the student to learn through exploration. Hao et al. (2022) induce a step-wise reward function from a language model trained in a supervised way. Building on this, Li et al. (2024a) apply RL to text generation KD, where a student model is trained by the REINFORCE algorithm (Williams, 1992) maximizing the cumulative reward suggested by the teacher. However, REINFORCE is known to suffer from high variance because it estimates gradient by sampled trajectories (i.e., sequences), which can vary significantly (Sutton & Barto, 2018). This issue is further exacerbated in text generation scenarios due to the large action space (i.e., vocabulary size), resulting in unstable learning.

In this paper, we propose BRIM, a novel **B**lock-wise **R**eturn **I**nduction **M**ethod for RL-based knowledge distillation. Our work is inspired by Li et al. (2024a), who derive a Q-value function from the teacher's policy (next-token probabilities) and induce a reward function based on the Bellman Optimality Equation (Bellman, 1952). In our approach, we break the long sampled trajectory of the student model into blocks of $K$ consecutive actions. For each block, we repeatedly apply the inverse of the Bellman Optimality Equation and induce a total reward for the block from the teacher model. Then, we sum the total rewards for all blocks as an approximate return (i.e., the total reward for

---

[1]Our code is released at https://anonymous.4open.science/r/BRIM-6070

a whole trajectory). We use such a block-wise approximated return as the RL training signal and update the student model with standard policy gradient (Sutton et al., 1999). Theoretical analysis shows that our BRIM reduces the variance of the total reward, thus effectively mitigating the high variance issue of RL-based text generation KD.

In essence, our approach is a REINFORCE-with-baseline (Williams, 1992; Sutton & Barto, 2018) variant that introduces an auxiliary term (called a *baseline*) to the return, which oftentimes stabilizes RL training (Sutton et al., 2000; Greensmith et al., 2004; Thomas & Brunskill, 2017). Traditional approaches have a baseline that solely depends on the sampled data (e.g., mean return of all samples), but this introduces additional noise to the training process if the sampled set is small or not representative. Later, researchers extend REINFORCE-with-baseline by developing Actor–Critic (AC) algorithms that learn a critic (i.e., estimated state-value function) to stabilize training. Our method also extends traditional REINFORCE-with-baseline but differs from AC: we derive a baseline term based on block-wise Bellman Optimality from the teacher model; thus, we do not need to train a cumbersome state-value function as AC algorithms do.

We evaluated our approach on three text generation datasets categorized into different domains: XSum (Narayan et al., 2018) for summarization, the Europarl corpora (Koehn, 2005) for machine translation, and GSM8K (Cobbe et al., 2021) for arithmetic reasoning. Experiments show that our proposed BRIM consistently achieves an add-on performance improvement when combined with the recent KD through the RL method (Li et al., 2024a). More importantly, we conduct an empirical analysis to show that our BRIM demonstrates lower variance and converges better than Li et al. (2024a), i.e., achieving a higher return and being more stable, which explains the observed improvements in empirical evaluation on downstream tasks.

## 2 METHODOLOGY

### 2.1 RL FORMULATION OF TEXT GENERATION

Text generation can be formulated as an undiscounted Markov Decision Process (MDP) with tuple $(\mathcal{S}, \mathcal{A}, T, r)$. The *state* space $\mathcal{S}$ includes all possible (sub)sequences and each of them is represented by $\mathbf{y}_{<t}$ for some time step $t$; notice that text generation may also depend on an input sequence, which is omitted here. The *action* $a_t \in \mathcal{A}$ at step $t$ corresponds to the next token $\mathbf{y}_t$ from the vocabulary $\mathcal{V}$. The *state transition* $T$ is a deterministic process in text generation, as $s_{t+1}$ is essentially the concatenation of $s_t$ and the newly generated word $a_t$. The *reward* function $r : \mathcal{S} \times \mathcal{A} \to \mathbb{R}$ provides feedback based on $(s_t, a_t)$. The goal of RL is to find a *policy* (distribution over actions) to maximize the expected *return* (cumulative rewards).

A key challenge in applying RL to text generation is the lack of well-defined step-wise reward functions. To address this, Hao et al. (2022) and Li et al. (2024a) assume that a language model generates the next word from a Boltzmann distribution based on the *Q-value function*,[2] given by

$$\pi_{\mathrm{LM}}(a \mid s) = \frac{\exp\big(q(s,a)\big)}{\sum_{a'} \exp\big(q(s,a')\big)}, \tag{1}$$

Due to the shared formula, a language model's pre-softmax logit can be viewed as the Q-value function, and with the Bellman optimality equation (Bellman, 1952), a step-wise reward function can be induced by

$$r(s_t, a_t) = q(s_t, a_t) - \max_{a' \in \mathcal{A}} q(s_{t+1}, a'). \tag{2}$$

Then, the goal of RL for text generation KD is to optimize the student's policy, denoted by $\pi_\theta$, to maximize the expected cumulative reward:

$$J(\theta) = \mathbb{E}_{\pi_\theta}\left[\sum_{t=1}^{T} r(s_t, a_t)\right], \tag{3}$$

---

[2]The Q-value function estimates the expected return (cumulative reward) of taking action $a$ in state $s$ and then following a given policy thereafter, defined by $q_\pi(s,a) = \mathbb{E}_\pi\left[\sum_{t=0}^{\infty} \gamma^t r_{t+1} \mid s_0 = s, a_0 = a\right]$.

The REINFORCE algorithm (Williams, 1992) is a policy gradient method, which is widely used for RL in NLP (Hao et al., 2022; Li et al., 2024a).

$$\nabla_\theta J(\theta) = \mathbb{E}_{\pi_\theta} \left[ \sum_{t=1}^{T} G_t \nabla_\theta \log \pi_\theta(a_t \mid s_t) \right] \tag{4}$$

where $G_t = \sum_{i=t}^{T} r(s_i, a_i)$ is a cumulative reward (i.e., return) from step $t$, and the expectation is approximated by Monte Carlo samples from the distribution $\pi_\theta$.

## 2.2 OUR BRIM METHOD

In this work, we address RL-based KD and propose to refine the learning signal $G_t$ in Eqn. (4) by extending the one-step reward induction to $K$ steps on a block-wise rollout sequence, which alleviates the high variance issue of RL. The key idea is to apply the inverse of the Bellman Optimality Equation for multiple steps, therefore directly connecting the Q-values at the current state with those of a future state.

We begin by considering the sum of rewards in Eqn. (2) over $K$ consecutive steps starting from step $t$, denoted by $G_{t:t+K}$:

$$
\begin{aligned}
G_{t:t+K} &:= \sum_{i=0}^{K-1} r(s_{t+i}, a_{t+i}) \\
&= \sum_{i=0}^{K-1} \left[ q(s_{t+i}, a_{t+i}) - \max_{a' \in \mathcal{A}} q(s_{t+i+1}, a') \right] \\
&= q(s_t, a_t) - \max_{a' \in \mathcal{A}} q(s_{t+K}, a')
\end{aligned} \tag{5}
$$

where Eqn. (5) assumes that an optimal action $a_{t+i+1} = \arg\max_{a' \in \mathcal{A}} q(s_{t+i+1}, a')$ is taken. However, a student's policy may not be optimal; therefore, Eqn. (5) becomes an approximation, denoted by $\hat{G}_{t:t+K}$,:

$$\hat{G}_{t:t+K} = q(s_t, a_t) - \max_{a' \in \mathcal{A}} q(\hat{s}_{t+K}, a') \tag{6}$$

where $\hat{s}_{t+K}$ is the state at the $(t+K)$th step after following the student's policy. This is a reasonable approximation because, in KD, a student is usually pretrained in a meaningful way (Turc et al., 2019; Lee et al., 2023; Kim et al., 2024) and the approximation will be more accurate as the optimization proceeds.

Building upon the $K$-step reward formulation, we can obtain an approximate return $\hat{G}_t$ by considering intervals of $K$ steps, i.e., $\hat{G}_{t:t+K}, \hat{G}_{t+K:t+2K}, \cdots$. Formally, we have

$$
\begin{aligned}
\hat{G}_t &= \sum_{i=0}^{\lfloor \frac{T-t+1}{K} \rfloor} \hat{G}_{t+iK:t+(i+1)K} \\
&= \sum_{i=0}^{\lfloor \frac{T-t+1}{K} \rfloor} \left[ q(s_{t+iK}, a_{t+iK}) - \max_{a' \in \mathcal{A}} q(\hat{s}_{t+(i+1)K}, a') \right].
\end{aligned} \tag{7}
$$

which will be used in our RL-based generation KD.

In particular, the student's policy is used to sample a sequence of actions (i.e., output words). Then, the sequence is fed to the teacher model, which evaluates the sequence by Eqn. (7). Finally, we follow the policy gradient formula, but use the approximate return for the update:

$$\nabla_\theta J(\theta) \approx \mathbb{E}_{\pi_\theta} \left[ \sum_{t=1}^{T} \hat{G}_t \nabla_\theta \log \pi_\theta(a_t \mid s_t) \right] \tag{8}$$

where $\hat{G}_t$ is our approximate return defined in Eqn. (7). The process is shown in Algorithm 1.

## 2.3 BIAS AND VARIANCE ANALYSIS

Although the REINFORCE algorithm (Williams, 1992) estimates gradients in an unbiased way, it is known to be noisy and prone to high variance in the gradient estimation, which may lead to instability in learning (Greensmith et al., 2004; Mnih et al., 2016; Bjorck et al., 2022).

A standard method to mitigate this issue is to subtract a *baseline* term $b_t$ from the actual return:

$$\hat{G}_t = G_t - b_t. \tag{9}$$

For example, the average return over a batch (Rosenberg, 2021) is commonly used as the baseline term to stabilize the REINFORCE algorithm.

Our BRIM approach is a variant of REINFORCE with baseline. This can be seen by examining the difference between the actual return $G_t$ and our approximate return $\hat{G}_t$. In our KD application, the actual return $G_t$ is given by accumulating the reward defined in Eqn. (2). In other words, we have

$$G_t = \sum_{i=0}^{T} \Big( q(s_{t+i}, a_{t+i}) - \max_{a' \in \mathcal{A}} q(s_{t+i+1}, a') \Big). \tag{10}$$

Combining Eqns. (7), (9), and (10), we can interpret our approximate return $\hat{G}_t$ as introducing a baseline term with the following form

$$b_t = \sum_{\substack{i=0 \\ i \not\equiv 0 \,(\mathrm{mod}\, k)}}^{T-1} \Big[ q(s_{t+Ki+1}, a_{t+Ki+1}) - \max_{a' \in \mathcal{A}} q(s_{t+Ki+1}, a') \Big]. \tag{11}$$

Unlike conventional, policy-independent baselines (Sutton & Barto, 2018; Rosenberg, 2021), our baseline depends on the selected actions and thus introduces bias into the expected return estimation. However, our approach can alleviate the high variance issue of REINFORCE with mild assumptions. The key insight is that Eqn. (5) cancels intermediate terms in the summation over different time steps, so the variance is reduced. This is formally analyzed by the following theorem.

**Theorem 1** (Variance Reduction via $K$-Step Return). *Let $G_t$ be the actual return and $\hat{G}_t$ be the $K$-step approximate return for some sequences sampled from the student policy $\pi$. Assuming that the state–action–reward tuples $(s_t, a_t, r_t)$ are iid drawn at different steps, we have:*

$$\mathrm{Var}[\hat{G}_t] \leq \mathrm{Var}[G_t]. \tag{12}$$

*Proof.* See Appendix B. □

The iid assumption is reasonable and widely adopted in theoretical RL research (Kearns & Singh, 2000; Bhandari et al., 2018; Xu et al., 2020), because in many environments the dependencies decay rapidly and correlation is further weakened when a large batch of samples is considered.

Overall, Theorem 1, along with the derivations in Appendix B, indicates that our BRIM alleviates variance at a power rate as $K$ increases, which is also empirically verified in §3.3. Although this method introduces a bias term in the gradient estimation, the bias is effectively mitigated: it diminishes for smaller values of $K$ and converges to zero as the student policy becomes more optimal. Detailed bias analysis is given in Appendix C. Such a trade-off is widely applied in existing RL literature, as seen in Temporal Difference (TD) learning (Sutton, 1988), Actor–Critic algorithms (Konda & Tsitsiklis, 1999; Mnih et al., 2016), and Deep Q-Network (DQN; Mnih et al., 2015).

## 3 EXPERIMENTS

In this section, we present the empirical evaluation and analysis of our proposed BRIM. We begin by describing the datasets, baseline methods, and implementation details, followed by the main results and detailed analyses.

## 3.1 SETTINGS

**Tasks, Datasets, and Metrics.** We evaluate our approach on various text generation tasks that are frequently considered in previous literature (Maruf et al., 2018; Magister et al., 2023; Wen et al., 2023; Touvron et al., 2023; Biderman et al., 2024; Wang et al., 2024).

- **XSum Summarization.** The Extreme Summarization (XSum) is a challenging dataset for text summarization introduced by Narayan et al. (2018), where the summaries are highly abstractive as they emphasize key ideas with novel wordings. We employ ROUGE[3] (Lin, 2004) as the primary metric, which is common practice in summarization (Ravaut et al., 2024; Van Veen et al., 2024; Agarwal et al., 2025).
- **Europarl EN–NL Translation.** Europarl (Koehn, 2005) is a high-quality, multilingual parallel corpus extracted from European Parliament proceedings. We choose English-to-Dutch, a relatively low-resource translation direction, to facilitate our distillation experiments. We report the BLEU score [3] (Papineni et al., 2002), character-level F score (chrF, Popović, 2015)[3], and translation edit rate (TER, Snover et al., 2006)[3], following the standard evaluation in machine translation (Barrault et al., 2019; Hrabal et al., 2024).
- **GSM8K Reasoning.** Grade School Math 8K (GSM8K, Cobbe et al., 2021) is a popular dataset consisting of around 8,000 grade school-level math problems with detailed step-by-step solutions. The standard evaluation metric for GSM8K is solution accuracy (Wang et al., 2024; Setlur et al., 2025), which is adopted in our experiments.

We employ the standard training, validation, and test splits for XSUM (Narayan et al., 2018) and Europarl (Koehn, 2005). For GSM8K, the standard split comprises only training and test sets (Cobbe et al., 2021). We adopt the open-source split provided by Wang et al. (2024), where the validation set is constructed by randomly selecting examples from the original training data.

**Implementation Details.** In our KD, the teacher is the 3B-parameter FLAN-T5-XL model (Chung et al., 2024), which shares the same architecture as prior work (Li et al., 2024a). For the summarization task, we directly prompt FLAN-T5-XL as it has already been instruction-finetuned for summarization. On the other tasks, FLAN-T5-XL yields subpar performance if prompted directly; we finetune the model as the teacher, which is commonly practiced in KD research (De Gibert et al., 2024; Setiawan, 2024; Ye et al., 2025).

The student uses the 250M-parameter T5-base model, which is consistent with the configuration in prior work Agarwal et al. (2024); Li et al. (2024a).

Following previous KD studies (Wen et al., 2023; Li et al., 2024a), we perform pre-distillation, where the student is pretrained by the cross-entropy loss based on the teacher's outputs. This ensures a meaningful initialization of the student model and enables effective exploration for reinforcement learning. Notice that text generation has a much larger state–action space than a typical RL environment such as Atari games (Mnih et al., 2015). The student performs greedy action selection when generating a sequence. Our return induction builds upon $K$-step Bellman optimality equations, and the hyperparameter $K$ is critical in our framework. We report performance for $K \in \{2, 4, 8, 16\}$ in our experiments.

Additional experimental details, including hyperparameter settings, statistical tests, and computing infrastructure, are provided in Appendix E.

**Competing Methods.** We compare our BRIM against both divergence-based and RL-based text generation KD:

- **KL Distillation** (Hinton et al., 2015). It minimizes the Kullback–Leibler (KL) divergence between student and teacher distributions.
- **SeqKD** (Kim & Rush, 2016). This is a classic method where the student maximizes the likelihood of teacher-generated sequences. It is a hard version of KL distillation
- **JS Distillation** (Wen et al., 2023). Jensen–Shannon (JS) divergence is a symmetric divergence that overcomes the over-smoothing problem of KL divergence (Wei et al., 2019).

---

[3]We computed ROUGE scores and BLEU scores using the implementation at google-research and sacrebleu, respectively. All ROUGE, BLEU, chrF, and TER scores are reported with a 95% confidence interval.

Figure 1: Average predicted return vs Approaches.

- **TVD Distillation** (Wen et al., 2023). The Total Variation Distance (TVD) is another symmetric divergence shown to outperform other methods (Wen et al., 2023). This method is also explored in Agarwal et al. (2024) with a ratio between the two terms of TVD.
- **LLMR** (Li et al., 2024a). In this method, a reward function is induced from a teacher language model by one-step Bellman optimality (Hao et al., 2022). Then, the student model is trained by RL towards the induced reward.

Since our approach reduces the variance of RL, we consider alternative variance reduction techniques under the LLMR framework:

- **LLMR + Mean Baseline.** Using the average reward in a batch as a baseline is commonly used for stabilizing RL training (Sutton & Barto, 2018).
- **LLMR + Min-Variance Baseline.** This is an advanced variant that is shown to be theoretically optimal when the baseline is derived from batch data (Rosenberg, 2021).

For a fair comparison, we apply the same settings in §3.1 (when applicable) to the competing methods as we do to our approach. Specifically, all methods adopt pre-distillation to ensure a meaningful student initialization, and all RL methods use the same action selection procedure.

### 3.2 MAIN RESULTS

As mentioned in §2.2, the primary advantage of our BRIM is its enhanced RL optimization compared with classic REINFORCE. In this part, we will first show that our approach indeed achieves a higher return (cumulative reward) in RL. Then, we will show that our approach leads to improved performance in NLP tasks.

**Return in RL.** The goal of RL is to learn a policy maximizing the expected return. Therefore, we may use it to evaluate the outcome of RL training.

Figure 1 shows the return score that is defined in Eqn. (10), where the return is averaged over different test samples, using various RL methods in the three NLP tasks. As seen, our BRIM consistently achieves a higher average return than competing approaches across all the tasks. This indicates that our BRIM learns a superior policy in terms of the return, which is precisely the RL objective.

In addition, we observe that an increased $K$ may not necessarily improve the return. This is because our BRIM introduces bias despite its reduced variance (§2.3). Therefore, a trade-off should be sought when choosing the $K$ value.

**NLP Task Performance.** Table 1 presents the results of our approach in NLP metrics.

We first examine the performance of directly prompting the teacher and the non-distilled student model in a zero-shot manner, offering empirical lower and upper bounds for the KD process. Note that the bounds are not theoretically guaranteed; instead, KD is empirically expected to improve the student's performance but may still underperform the teacher, especially when the student is small. In our setup, the student is a T5-base model, which does not yield reasonable performance when prompted directly.

We then consider divergence-based distillation methods, including SeqKD and KL/JS/TVD distillations. As seen from the table, symmetric methods (JS, TVD)—which involve both exploitation of teacher predictions and exploration based on student predictions—tend to surpass asymmetric methods (SeqKD, KL), where the student follows teacher predictions without any exploration. The results are consistent with previous findings (Wen et al., 2023; Agarwal et al., 2024).

| Model | | XSum | | | Europarl | | | GSM8K |
|---|---|---|---|---|---|---|---|---|
| | | ROUGE-1$^\uparrow$ | ROUGE-2$^\uparrow$ | ROUGE-L$^\uparrow$ | BLEU4$^\uparrow$ | chrF$^\uparrow$ | TER$^\downarrow$ | Accuracy(%)$^\uparrow$ |
| Teacher | | 41.32 | 18.86 | 33.79 | 25.36 | 51.11 | 63.17 | 40.71 |
| Student | | 19.60 | 3.19 | 13.72 | 0.95 | 24.80 | 100.21 | 0.00 |
| Distilled Student | SeqKD Kim & Rush (2016) | 33.54 | 11.90 | 26.67 | 22.09 | 48.33 | 66.18 | 20.02 |
| | KL Hinton et al. (2015) | 34.36 | 12.86 | 27.38 | 22.35 | 48.58 | 65.93 | 23.96 |
| | JS Wen et al. (2023) | 34.87 | 13.18 | 27.84 | 22.55 | 48.71 | 65.74 | 24.72 |
| | TVD Wen et al. (2023) | 35.17 | 13.30 | 28.10 | 22.63 | 48.66 | 65.79 | 24.94 |
| | LLMR Li et al. (2024a) | 35.54 | 13.70 | 28.56 | 22.72 | 49.04 | 65.38 | 25.21 |
| | LLMR + Mean baseline | 35.60 | 13.76 | 28.64 | 22.67 | 49.03 | 65.39 | 25.39 |
| | LLMR + Min-Var baseline | 35.59 | 13.78 | 28.66 | 22.70 | 48.97 | 65.55 | 25.10 |
| | BRIM ($K = 2$) | **36.63** | 14.15 | **29.29** | 22.93 | **49.25** | **65.15** | 25.63 |
| | BRIM ($K = 4$) | 36.42 | **14.16** | 29.08 | 22.93 | 49.21 | 65.21 | 25.93 |
| | BRIM ($K = 8$) | 35.68 | 13.88 | 28.76 | **22.95** | 49.23 | 65.20 | **26.38** |
| | BRIM ($K = 16$) | 35.31 | 13.68 | 28.51 | 22.94 | 49.24 | 65.18 | 26.16 |

Table 1: Main results on XSum, Europarl EN–NL, and GSM8K datasets. The best student result is in **bold**. $^{\uparrow/\downarrow}$The higher/lower, the better. We prompt the teacher and off-the-shelf student in a zero-shot manner to gain the first two rows.

Next, we evaluate LLMR (Li et al., 2024a), a text generation KD approach using REINFORCE. Results show that LLMR provides certain performance gain over non-RL KD methods, which is likely stemmed from the student's self-exploration, aligning with the observations in Li et al. (2024a) and other recent RL-based text generation research (Ouyang et al., 2022; DeepSeek-AI et al., 2025).

To mitigate the high variance of REINFORCE in LLMR, we incorporate classic RL baseline terms (mean baseline and min-variance baseline) that are estimated from batch data. However, these methods are not effective in our scenario, as text generation has a very large state–action space, which makes the generated outputs in a batch less representative and the baseline term less useful.

By contrast, our BRIM employs a novel baseline formulation that largely reduces the variance of RL (Theorem 1) and improves RL optimization (Figure 1). Consequently, it delivers a noteworthy add-on performance gain on top of LLMR across three text generation tasks.

In the experiment, we also observe that a moderate $K$ between 2 to 8 leads to the highest NLP performance, which is consistent with the return analysis in Figure 1. It is also noticed that RL return and NLP performance are not perfectly correlated, as the induced reward may not fully reflect the task metric such as BLEU and ROUGE scores, which is also known as reward hacking (Amodei et al., 2016; Hao et al., 2022; Ouyang et al., 2022).

**Summary.** Our main results show that our BRIM (with a moderate $K$) improves RL optimization, which is generally translated to higher performance in various NLP tasks.

### 3.3 IN-DEPTH ANALYSES

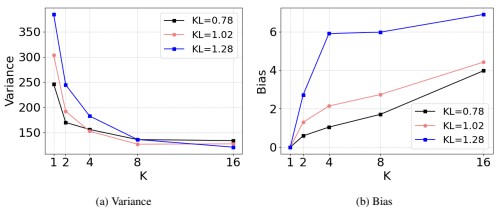

(a) Variance    (b) Bias

Table 2: Variance and bias with different $K$.

**Variance and bias analysis.** As shown by the theoretical analysis in §2.3, our approach provides a bias–variance trade-off by largely reducing the variance, although introducing a bias term. We empirically verify them in this analysis.

Figure 2a shows the variance of the $K$-step return, where we sample 32 outputs for a given input and use Eqn. (20) to estimate the variance of return; the variance is further averaged over 10K input samples. For the bias, we use Eqn. (25) for empirical estimation, and the results are shown in Figure 2b. We choose the value of $K$ from $\{1, 2, 4, 8, 16\}$ to see the trends. Note that $K = 1$ corresponds to the competing approach LLMR (Li et al., 2024a). In addition, we examine the impact of the initial student policy by considering students with various KL divergence levels from the teacher policy: a smaller KL divergence indicates that the student and teacher are more resemblant.

We observe that the variance decreases drastically as $K$ increases, while the bias term increases steadily. The observations align with our theoretical analysis in §2.3 and Appendix C, suggesting the need for seeking a moderate $K$ value to balance bias and variance.[4]

---

[4]Our bias–variance trade-off is different from that in a regression analysis (Hastie et al., 2009; Vapnik, 2013), where the total squared error is the sum of variance and squared bias, plus an irreducible noise. By contrast, the variance of return affects the smoothness of RL training, while bias affects the optimum quality (if converging); their total effect is not given by a simple addition.

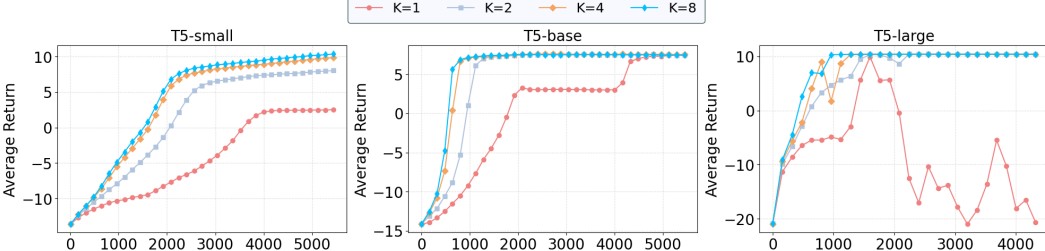

Figure 2: Learning curves. $y$-axis is the true return value, and $x$-axis is the number of training steps.

We also observe that when the student policy is initialized closer to the teacher policy (i.e., a smaller KL divergence), our BRIM generally demonstrates lower bias and variance. The bias reduction is predicted by our theoretical analysis in Appendix C, whereas the variance reduction is an empirical observation. Overall, the results demonstrate that pre-distillation is important to RL training for text generation, which is consistent with previous work (Ouyang et al., 2022; DeepSeek-AI et al., 2025).

**Model Size.** We analyze RL-based KD approaches with different student sizes. Figure 2 presents the learning curves for student models initialized from FLAN-T5-small (77M), FLAN-T5-base (250M), and FLAN-T5-large (800M) using our BRIM and the competing approach LLMR ($K = 1$).

As seen from the curves in Figure 2, LLMR exhibits notable instability during RL training as the model size increases, especially when scaling to FLAN-T5-large. Such a phenomenon is also reported in the RL literature: a large network is prone to overfit the limited sampled outputs, consequently leading to unstable performance on test data (Henderson et al., 2018; Cobbe et al., 2019).

In contrast, our BRIM largely alleviates this issue by reducing the variance, which stabilizes the learning curves. Overall, our method achieves smoother training and higher performance with all model sizes, compared with LLMR.

**LLM Evaluation.** We conduct an LLM evaluation as a surrogate of human evaluation, as classic NLP metrics (such as ROUGE and BLEU) may not fully reflect the quality of generated text. Specifically, we prompt the `Qwen2.5-72B-Instruct` (Qwen et al., 2025) LLM to conduct a pairwise evaluation of system outputs, against the commonly used KL distillation. We select TVD, LLMR, and our BRIM from Table 1 as the competitors, as pairwise evaluation is expensive. Our LLM evaluation considers multiple criteria, including overall quality, informativeness, and coherence. For each comparison, we query the LLM four times by swapping the two candidates and their IDs (namely, A and B), as LLM is prone to ID bias (Zheng et al., 2023) and positional bias (Shen et al., 2023). The detailed prompts are presented in Appendix H.

| Dataset | Method | Overall | Informativeness | Coherence |
|---|---|---|---|---|
| XSum | TVD | 67.50% | 68.15% | 65.90% |
| | LLMR | 69.95% | 70.55% | 66.30% |
| | BRIM | **73.50%** | **73.90%** | **70.40%** |
| Europarl | TVD | 53.80% | 54.15% | 54.85% |
| | LLMR | 56.45% | 55.85% | 56.30% |
| | BRIM | **58.85%** | **57.95%** | **58.45%** |

Table 3: LLM-based evaluation.

Table 3 shows the results of the LLM evaluation. We observe that our BRIM achieves the best winning rate in terms of all criteria (overall quality, informativeness, and coherence) on both datasets. These compelling results are consistent with the traditional task metrics in Table 1 and further demonstrate the effectiveness of our BRIM.

## 4 RELATED WORK

**Knowledge Distillation.** The foundation of KD is laid by Buciluǎ et al. (2006), who performs KD by aligning the logits of the student with those of a teacher through squared error minimization. This framework is extended by Hinton et al. (2015), who propose to use KL divergence to match the output probability distributions of the teacher and student. Kim & Rush (2016) extend KD to the sequence level for auto-regressive models, and Wen et al. (2023) further propose a general framework of $f$-divergence minimization to mitigate the mode averaging and collapsing issues. Agarwal et al. (2024); Gu et al. (2024); Ko et al. (2024); Wu et al. (2025) extends or refines the $f$-divergence minimization framework, enabling more effective or efficient training of the student model. These divergence-based KD approaches heavily rely on imitation of the teacher's predictions, neglecting the student's active exploration during learning.

Other distillation methods are less comparable as they target specialized scenarios, such as specific data settings (Zhao et al., 2024; Zhou et al., 2024; Liu et al., 2024b), distinct downstream tasks (Liu

et al., 2024a; Zhang et al., 2024a), or architectural variations (Zhang et al., 2024b; Chen et al., 2024; Peng & Zhang, 2025). In contrast, our work focuses on the general task of text generation, so these approaches are not directly related or comparable to ours.

**Bridging RL and text generation KD.** Recent work has sought to combine RL and KD by deriving rewards from teacher models. Hao et al. (2022) interpret a supervised-trained language model's pre-softmax logits as Q-values, deriving a step-wise reward function via Bellman Optimality equation, which alleviates the sparse reward issue commonly existing in other RL text generation scenarios (Wu et al., 2018; Ouyang et al., 2022). Building on this, Li et al. (2024a) extend this approach to KD settings, where they induce a reward function from a large language model (serves as a teacher) and train a student model to maximize the teacher-induced return. However, RL is known to suffer from high variance, and our paper proposes BRIM that largely reduces the variance of RL training.

**Variance Reduction in RL.** REINFORCE with baseline (Sutton & Barto, 2018) mitigates the high variance issue by subtracting a baseline term derived from batch data. Actor–Critic methods (Konda & Tsitsiklis, 1999; Mnih et al., 2016) address this by learning a value function (critic) as the baseline term, but the inaccurate value estimates from the critic can lead to harmful updates in the actor's policy, while a poor decision by the actor can adversely affect the critic's learning. This often results in divergence of RL training (Bhatnagar et al., 2007; Fujimoto et al., 2018; Parisi et al., 2019). Thus, recent RL work for LLMs tends to avoid learning a critic (DeepSeek-AI et al., 2025). Our BRIM exploits the mathematical structure of LM-induced rewards to derive a principled baseline for variance reduction, without learning an auxiliary neural network like a critic.

Another line of studies develops conservative policy optimization techniques like TRPO (Schulman et al., 2015a) and PPO (Schulman et al., 2017). Appendix D presents empirical results showing that BRIM integrates cleanly into PPO by replacing the critic-based advantage with a $K$-step return (§2.2). This removes the need to learn a critic, mitigating error accumulation in critic and bias in advantage estimates, and yields stable, near-monotonic return improvements (Figure 3b). By contrast, coupling PPO with the competing approach LLMR introduces compounding critic errors and biased advantages, leading to unstable updates and training collapse (Figure 3a and Figure 3b), which further underscores the advantage of BRIM.

$N$**-Step Bootstrapping.** In value-based RL, the state-value function can be estimated either from one-step TD bootstrapping ($N = 1$) or from the full Monte Carlo return ($N \to \infty$). Classical $N$-step TD interpolates between these two extremes. Although this appears related to our method, the $N$-step formulation is fundamentally different from our BRIM. First, the symbols $N$ and $K$ encode distinct mathematical concepts: $N$ specifies the rollout length of the TD target, whereas $K$ denotes the depth of the inverse Bellman expansion that defines our variance-reduction baseline. Second, $N$-step TD requires learning an additional parameterized state-value function, which is impractical in our setting. Thus, algorithms built on $N$-step bootstrapping, including Eligibility Trace (Sutton, 1988) and its modern variants (Schulman et al., 2015b), lie outside the scope of this work.

**Other RL-based KD approaches.** Recent RL-based knowledge distillation methods often adapt the Ouyang et al. (2022)'s framework to train reward functions use Bradley–Terry model (Gu et al., 2025; Li et al., 2025; Nath & coauthors, 2025; Gao et al., 2025; Zhang et al., 2024a) or its variants (Zhang et al., 2025; Li et al., 2024b). Alternatively, some approaches use rule-based rewards for tasks like reasoning (Xu et al., 2025a;b). However, these methods rely on sequence-level rewards, which are sparse and result in poor credit assignment. In contrast, our approach induces block-wise reward estimation to provide a denser, more granular training signal, offering a fundamental training advantage.

## 5 CONCLUSION

In this paper, we introduce BRIM, a novel return induction via applying the inverse of the Bellman Optimality Equation to the block-wise sampled trajectory for reinforcement learning for knowledge distillation in the text generation domain. Compared with conventional RL methods, our approach effectively reduces gradient variance and leads to a more stable training, shown by both theoretical and empirical analyses. Extensive experiments across diverse text generation tasks verify that our approach improves RL training and boosts NLP task performance.

## 6 LIMITATIONS AND FUTURE WORK

A limitation of our evaluation is that we use LLM-as-judge as a surrogate for human evaluation. Although LLM-as-judge has become a scalable and common approach, it can possess intrinsic biases or fail to capture the true human preferences regarding text quality. To address this, we take specific measures, e.g., enumerating all combinations of answer IDs and order, to mitigate position and ID bias, which improves the trustworthiness of the LLM-as-judge assessment.

Our current method uses a fixed hyperparameter $K$, which is tuned by validation. In future work, we will explore an adaptive $K$-step estimation method, which would dynamically adjust the block size, potentially based on the student's current performance or policy uncertainty. This could better manage the exploration–exploitation trade-off and reduce the need for hyperparameter tuning.

In addition, our current scope is inner-family distillation, where the teacher and student share a common tokenizer and vocabulary. For cross-family distillation, there is a significant challenge that we need to map the tokenizers of different models. While tackling this token mapping problem is an important research direction, it goes beyond the scope of this paper and we are happy to explore it in future work.

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

## A  THE USE OF LARGE LANGUAGE MODELS (LLMS)

We employ Large Language Models (LLMs) as general-purpose tools to improve writing quality, including grammar, spelling, and sentence structure. In addition, we use LLMs to refine LaTeX syntax and formatting.

## B  PROOF OF THEOREM 1

Using $K$-step returns as a learning signal to learn a student policy $\pi$ guarantees reduced variance in return estimation compared to the full trajectory return, i.e., $\text{Var}[\hat{G}_t] \leq \text{Var}[G_t]$. (Detailed in Theorem 1).

*Proof.* We denote the variance of $q(s,a)$ and $\max_{a' \in \mathcal{A}} q(s,a')$ as:

$$\sigma_{\mathcal{S},\mathcal{A}}^2 = \text{Var}_{s,a}\big[q(s,a)\big], \tag{13}$$

$$\sigma_{\mathcal{S}}^2 = \text{Var}_s\Big[\max_{a' \in \mathcal{A}} q(s,a')\Big]. \tag{14}$$

We first decompose the variance of the actual return $G_t$:

$$\text{Var}[G_t] = \text{Var}\bigg[\sum_{i=0}^{T-t} r_{t+i}\bigg] \qquad \text{[definition of } G_t] \tag{15}$$

$$= \sum_{i=0}^{T-t} \text{Var}\Big[q(s_{t+i}, a_{t+i}) - \max_{a' \in \mathcal{A}} q(s_{t+i+1}, a')\Big] \qquad \text{[iid assumption]} \tag{16}$$

$$= \sum_{i=0}^{T-t} \Big(\text{Var}\big[q(s_{t+i}, a_{t+i})\big] + \text{Var}\big[\max_{a' \in \mathcal{A}} q(s_{t+i+1}, a')\big]\Big) \qquad \text{[iid assumption]} \tag{17}$$

$$= \sum_{i=0}^{T-t} \big(\sigma_{\mathcal{S},\mathcal{A}}^2 + \sigma_{\mathcal{S}}^2\big) \tag{18}$$

$$= (T - t + 1)\big(\sigma_{\mathcal{S},\mathcal{A}}^2 + \sigma_{\mathcal{S}}^2\big). \tag{19}$$

Next, we decompose the variance of our $K$-step approximate return $\hat{G}_t$:

$$\text{Var}[\hat{G}_t] = \text{Var}\left[\sum_{i=0}^{\lfloor\frac{T-t}{k}\rfloor}\left(q(s_{t+ik}, a_{t+ik}) - \max_{a'\in\mathcal{A}} q(s_{t+(i+1)k}, a')\right)\right] \qquad \text{[by Eqn.( 7) in the main text]}$$

$$(20)$$

$$= \sum_{i=0}^{\lfloor\frac{T-t}{k}\rfloor} \text{Var}\left[q(s_{t+ik}, a_{t+ik}) - \max_{a'\in\mathcal{A}} q(s_{t+(i+1)k}, a')\right] \qquad \text{[iid assumption]}$$

$$(21)$$

$$= \sum_{i=0}^{\lfloor\frac{T-t}{k}\rfloor} \left(\text{Var}\left[q(s_{t+ik}, a_{t+ik})\right] + \text{Var}\left[\max_{a'\in\mathcal{A}} q(s_{t+(i+1)k}, a')\right]\right) \qquad \text{[iid assumption]}$$

$$(22)$$

$$= \sum_{i=0}^{\lfloor\frac{T-t}{k}\rfloor} \left(\sigma_{\mathcal{S},\mathcal{A}}^2 + \sigma_{\mathcal{S}}^2\right) \qquad (23)$$

$$= \left(\left\lfloor\frac{T-t}{k}\right\rfloor + 1\right)\left(\sigma_{\mathcal{S},\mathcal{A}}^2 + \sigma_{\mathcal{S}}^2\right). \qquad (24)$$

Comparing Eqns. (19) and (24), we immediately have $\text{Var}[\hat{G}_t] \leq \text{Var}[G_t]$, completing the proof. $\qquad\square$

## C  BIAS ANALYSIS

In this section, we analyze the bias introduced by using the $K$-step return $\hat{G}_t$ in place of the actual return $G_t$. Recall that they differ by a baseline term shown in Eqns. (9) and (11) in the main text, and this discrepancy introduces bias in the return estimation:

$$\text{bias of return} = \mathbb{E}_{\pi_\theta}\left[(\hat{G}_t - G_t)\right] = \mathbb{E}_{\pi_\theta}\left[\sum_{\substack{i=0 \\ i\not\equiv 0\,(\text{mod}\,k)}}^{T-1}\left[q(s_{t+Ki+1}, a_{t+Ki+1}) - \max_{a'\in\mathcal{A}} q(s_{t+Ki+1}, a')\right]\right]$$

$$(25)$$

gradient estimation:

$$\text{bias of gradient} = \mathbb{E}_{\pi_\theta}\left[(\hat{G}_t - G_t)\nabla_\theta \log\pi_\theta(a_t \mid s_t)\right] = \mathbb{E}_{\pi_\theta}\left[-b_t\nabla_\theta \log\pi_\theta(a_t \mid s_t)\right] \qquad (26)$$

We show below that a smaller value of $K$ reduces bias, providing a bias-variance tradeoff for REIN-FORCE. Further, we will show that the bias converges to zero as the student policy becomes more optimal, assuming all Q-values are distinct.

**Bias Reduction with Smaller $K$.**  The baseline term defined in Eqn. (11) in the main text is given by

$$b_t = \sum_{\substack{i=0 \\ i\not\equiv 0\,(\text{mod}\,k)}}^{T-1}\left[q(s_{t+Ki+1}, a_{t+Ki+1}) - \max_{a'\in\mathcal{A}} q(s_{t+Ki+1}, a')\right]. \qquad (27)$$

Since

$$q(s_{t+Ki+1}, a_{t+Ki+1}) - \max_{a'\in\mathcal{A}} q(s_{t+Ki+1}, a') \leq 0, \qquad (28)$$

a smaller $K$ reduced the number of terms in the summation. This decreases $|b_t|$, which in turn decreases the magnitude of the gradient bias in Eqn. (26).

---

**Algorithm 1** BRIM

---

**Input:** Non-parallel dataset $D$; teacher Q-value function $q : \mathcal{S} \times \mathcal{A} \to \mathbb{R}$; student policy $\pi_\theta$ with initial parameters $\theta$; segment length $K$; learning rate $\eta$; maximum rollout length $T$; number of iterations $U$

**Output:** Trained student policy $\pi_\theta$

**for** $j \leftarrow 1$ **to** $U$ **do**

    Sample a source sentence $\mathbf{x} \in D$

    Set the initial state $s_0 \leftarrow \mathbf{x}$

    Generate a trajectory $\tau = \{(s_0, a_0), (s_1, a_1), \ldots, (s_T, a_T)\}$ by sampling from $\pi_\theta$

    Initialize gradient accumulator: $g \leftarrow 0$

    **for** $t \leftarrow T$ **to** $0$ **do**

      **if** $t = T$ **then**

        $\hat{G}_T \leftarrow q(s_T, a_T)$

      **else if** $T - t < k$ **then**

        $\hat{G}_t \leftarrow \left[ q(s_t, a_t) - \max_{a' \in \mathcal{A}} q(s_{t+1}, a') \right] + \hat{G}_{t+1}$

      **else**

        $\hat{G}_t \leftarrow \left[ q(s_t, a_t) - \max_{a' \in \mathcal{A}} q(s_{t+K}, a') \right] + \hat{G}_{t+K}$

      **end**

      $g \leftarrow g + \hat{G}_t \nabla_\theta \log \pi_\theta(a_t \mid s_t)$

    **end**

    $\theta \leftarrow \theta + \eta\, g$

**end**

**return** $\pi_\theta$

---

**Bias Convergence to Zero.** Suppose the student policy is optimal, i.e., greedy with respect to the teacher's Q-value function $q(s, a)$, given by

$$a_{t+i} = \arg\max_{a' \in \mathcal{A}} q(s_{t+i}, a'). \tag{29}$$

It is easy to see from Eqn. (27) that $b_t = 0$, implying that

$$\mathbb{E}_{\pi_\theta} \left[ b_t \nabla_\theta \log \pi_\theta(a_t \mid s_t) \right] = 0. \tag{30}$$

Suppose the Q-values for different actions are distinct (in which case $\mathrm{argmax}$ is continuous), the result further suggests that the bias term would converge to zero, if the student policy is closer to optimal during training.

# D  PPO IN RL-BASED TEXT GENERATION KD

We run PPO-based training experiments with both LLMR and our BRIM in the RL-based text generation KD setting (§2.1) to assess whether PPO's learning framework can be productively incorporated into this scenario.

**PPO with LLMR.** As a competing approach, we run PPO where the reward signal for policy learning is the teacher-induced reward in Eqn. (2). Our PPO pipeline follows the setup of Ouyang et al. (2022): we learn a state-value function by minimizing the TD error (i.e., making the earlier value prediction agree with a later, better-informed bootstrapped return to enforce temporal consistency), and we update the policy using the critic's advantage estimates as a baseline (following (Schulman et al., 2015b)), increasing the likelihood of actions with positive advantage and decreasing it for those with negative advantage.

**PPO with BRIM.** We then incorporate PPO into BRIM by removing all learning components related to the critic value function and replacing the critic's advantage used in LLMR's PPO setting with our block-wise $K$-step return estimator introduced in §2.2.

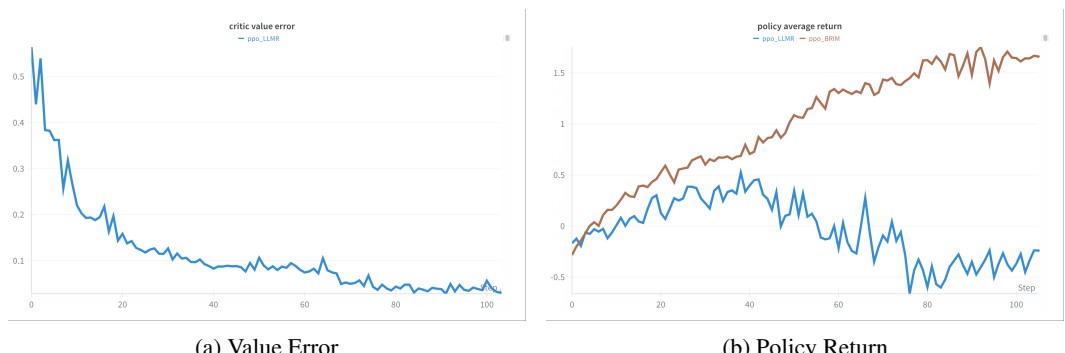

           (a) Value Error                         (b) Policy Return

Figure 3: PPO training with LLMR

| Dataset | Task | # of Samples | | |
|---|---|---|---|---|
| | | Train | Dev | Test |
| XSum (Narayan et al., 2018) | Summarization | 202,926 | 11,332 | 11,333 |
| Europarl EN-NL (Koehn, 2005) | Machine Translation | 1,167,808 | 10,014 | 10,016 |
| GSM8K (Cobbe et al., 2021) | Arithmetic reasoning | 6,705 | 768 | 1,319 |

Table 4: Statistics of our datasets.

**Results.** With LLMR, PPO fails to learn a useful policy: the critic's TD error (Fig. 3a) converges to a small value, while the average policy return (Fig. 3b) diverges, indicating that PPO does not discover a stable, high-reward strategy in this setup. In contrast, under BRIM, the average policy return in Fig. 3b exhibits a nearly monotonic increase, suggesting that BRIM effectively resolves the training-instability issues observed with LLMR.

**Discussion.** The instability of LLMR under PPO primarily stems from inaccuracies in the learned critic. In PPO with LLMR, two coupled error sources undermine learning: (i) inaccuracies in the learned critic which stemming from bootstrapping errors that accumulate over time, and (ii) biased advantage estimates (e.g., from GAE with $\lambda < 1$) that are computed on top of those inaccurate value predictions. When combined within PPO's update, these errors distort both the baseline and the policy gradient, so the policy is optimized toward a mis-specified training signal, leading to instability and, empirically, divergence of the return curve. Similar fragility of value-learning has been reported in recent RL/NLP work (Moalla et al., 2024; Yuan et al., 2025), motivating methods that avoid an auxiliary critic (DeepSeek-AI et al., 2025). Following this direction, BRIM replaces the critic with a teacher-induced block-wise $K$-step return, which serves as a stable surrogate for advantage computation, mitigates error accumulation, and yields more reliable policy improvement.

## E  EXPERIMENTAL SETTING DETAILS

**Computing Infrastructure.** Experiments were conducted on a Linux server equipped with an AMD EPYC 7313 CPU (32 GB RAM) and an NVIDIA RTX A6000 GPU (48 GB VRAM). The system uses NVIDIA driver v560.28.03 and CUDA Toolkit 12.6 (as reported by `nvidia-smi`). Software and dependency versions are listed in the `requirements.txt` file of our anonymous GitHub repository: https://anonymous.4open.science/r/BRIM-6070.

**Hyperparameter Settings.** For our RL-based distillation experiments, we follow the configuration of Li et al. (2024a), employing the AdamW optimizer (Loshchilov & Hutter, 2019) with default parameters ($\beta_1 = 0.9$, $\beta_2 = 0.999$). All other hyperparameters—batch size, gradient accumulation steps, reward clipping range, dropout rate, warmup steps, and learning rate—are identical to those in (Li et al., 2024a). Since our datasets differ, we adjust the maximum input and output lengths for each text-generation task according to the recommendations of (Wen et al., 2023; Wang et al., 2024). Detailed settings are provided in Table 5.

For divergence-based KD competing approaches, we adopt the hyperparameter configurations from (Wen et al., 2023). The specific values for each parameter are summarized in Table 6.

**Statistical Analysis.** To quantify the variance of each approach we repeat every training procedure $N = 5$ times, using distinct random seeds drawn uniformly from $([1, 100\,000])$: $\{19083, 34007, 84122, 310, 55080\}$. For every run, we select the checkpoint that achieves the best validation performance and report its corresponding test-set score. We adopt the standard train/validation/test splits for XSUM and EUROPARL. Because the official GSM8K release lacks a validation split, we use the public split of Wang et al. (2024). Statistical significance between our method BRIM and each baseline is assessed with a paired two-sided $t$-test over the five seeds; $p < 0.035$ indicating the differences are deemed significant.

| Hyperparameter | Value |
| --- | --- |
| Training Epochs | 3 |
| Train Batch size | 8 |
| Eval Batch size | 32 |
| Optimizer | AdamW |
| Grad Accumulation Steps | 32 |
| Reward Clip Range | [-100, 100] |
| Dropout | 0.0 |
| Warmup Steps | 5,000 |
| Warmup Schedule Linear | (from 0 to LR) |
| Learning Rate (LR) | 0.00001 |
| Max Input Length | 1024 (Xsum) / 80 (Europarl) / 200 (GSM8K) |
| Max Output Length | 64 (Xsum) / 80 (Europarl) / 300 (GSM8K) |

Table 5: Hyperparameter Details for experiments on RL-based approaches (BRIM, LLMR, LLMR with mean baseline, and LLMR with min-variance baseline).

| Hyperparameter | Value |
| --- | --- |
| Training Epochs | 2 |
| Train Batch size | 32 |
| Eval Batch size | 32 |
| Optimizer | AdamW |
| Grad Accumulation Steps | 16 |
| Dropout | 0.25 |
| Warmup Steps | 5,000 |
| Warmup Schedule Linear | (from 0 to LR) |
| Learning Rate (LR) | 0.00005 |
| Max Input Length | 1024 (Xsum) / 80 (Europarl) / 200 (GSM8K) |
| Max Output Length | 64 (Xsum) / 80 (Europarl) / 300 (GSM8K) |

Table 6: Hyperparameter Details for experiments on divergence-based KD approaches (seqKD, KL, JSD, TVD).

# F  RESULTS ON MORE MODELS

KD studies on seq2seq tasks have largely centred on encoder-decoder structures such as T5 (Raffel et al., 2020; Chung et al., 2024) and BART (Lewis et al., 2020) models (Wen et al., 2023; Li et al., 2024a; Agarwal et al., 2024; Jung et al., 2024; Wang et al., 2025). To answer reviewers' likely question about BRIM's performance on recent popular decoder-only architectures, we also applied it to the Qwen1.5 model series (Qwen-Team, 2024) and report the results in Table 7.

| Model | XSum (ROUGE-1↑) | Europarl (BLEU4↑) | GSM8K (Acc. (%)↑) |
|---|---|---|---|
| Teacher (Qwen1.5-4B) | 38.15 | 21.32 | 42.08 |
| Student (Qwen1.5-0.5B) | 8.80 | 0.02 | 0.00 |
| KL  Hinton et al. (2015) | 31.29 | 15.76 | 26.31 |
| TVD  Wen et al. (2023) | 31.18 | 16.22 | 26.99 |
| LLMR  Li et al. (2024a) | 31.61 | 15.90 | 27.29 |
| BRIM | **32.28** | **16.46** | **28.13** |

Table 7: Distillation results on XSum, Europarl EN–NL, and GSM8K using Qwen1.5 models. Higher ↑ is better. The best $K$ values are 2, 2, and 16 for the three datasets, respectively.

| Models | XSum (max_output_len=64) | Europarl (max_output_len=64) | GSM8K (max_output_len=256) |
|---|---|---|---|
| T5 optimal $K$ | 2 | 2 | 8 |
| Qwen optimal $K$ | 2 | 2 | 16 |

Table 8: Summarization of optimal $K$ values for T5 and Qwen models on XSum, Europarl EN–NL, and GSM8K datasets.

## G   HYPERPARAMETER $K$ DISCUSSION

$K$ **and the exploration horizon**   Eqn. (7) in the main text shows that the $K$-step return depends on the rollout horizon $T$ and the truncation parameter $K$. With $T$ fixed (reflected by the maximum output length in the text generation scenario), the approximation error is governed solely by $K$. Table 8 indicates that the same $K$ performs robustly across tasks that share an identical horizon.

**A single optimal $K$ is elusive**   Because tasks differ in their typical horizons (output lengths), the $K$ that is optimal for short summaries may be sub-optimal for long-form reasoning. Adapting $K$ on a per-example basis would be ideal, but is infeasible in training with batch implementation. Instead, we conduct a lightweight grid search over $\{2, 4, 8, 16\}$ for each dataset and select the empirically best value.

## H   PROMPTS TEMPLATES FOR LLM EVALUATION

Table 9 and Table 10 present our prompts template for LLM evaluation on the summarization task and machine translation task, respectively.

## I   THE USE OF LARGE LANGUAGE MODELS (LLMS)

Gemini2.5 Comanici et al. (2025) was used in a limited capacity to improve writing quality, including checking grammar and rephrasing certain expressions with better sentence structures. In addition, we use it for formatting LaTeX tables and Matplotlib figures. However, we came up with the research ideas, conducted the analyses, and presented the contents without using AI tools.

## J   REPRODUCIBILITY STATEMENT

All code is provided via an anonymized Github Repository, including implementations for data loading, reward model training, and policy optimization. The datasets used are publicly available, and we release the complete set of training hyperparameters. Our evaluation approaches are also publicly available and can be fully reproduced.

Please evaluate the overall quality of the following summaries given the document.

Evaluation Criteria:
Overall Quality: A good summary should be both precise and concise, summarizing the most important points in the given document, without including unimportant or irrelevant details

Document: **[Source]**
Summary **[ID1]**: **[Summary-A]**
Summary **[ID2]**: **[Summary-B]**

FIRST, provide a one-sentence comparison of the two summaries for overall quality, explaining which you prefer and why.
SECOND, on a new line, state only the ID to indicate your choice. Your response should use the format:
Overall Quality: ¡one-sentence comparison and explanation¿
Preferred: ¡summary ID¿

Please evaluate the informativeness of the following summaries given the document.

Evaluation Criteria:
Informativeness: Does it include the most important details while excluding irrelevant content?

Document: **[Source]**
Summary **[ID1]**: **[Summary-A]**
Summary **[ID2]**: **[Summary-B]**

FIRST, provide a one-sentence comparison of the two summaries for informativenss, explaining which you prefer and why.
SECOND, on a new line, state only the ID to indicate your choice. Your response should use the format:
Informativeness: ¡one-sentence comparison and explanation¿
Preferred: ¡summary ID¿

Please evaluate the coherence of the following summaries given the document.

Evaluation Criteria:
Coherence: Is the summary logically structured and easy to follow?

Document: **[Source]**
Summary **[ID1]**: **[Summary-A]**
Summary **[ID2]**: **[Summary-B]**

FIRST, provide a one-sentence comparison of the two summaries for coherence, explaining which you prefer and why.
SECOND, on a new line, state only the ID to indicate your choice. Your response should use the format:
Informativeness: ¡one-sentence comparison and explanation¿
Preferred: ¡summary ID¿

Table 9: Prompt templates for LLM evaluation on the summarization task in terms of overall quality, informativeness, and coherence. Here, "**Source**" is the document to be summarized. The choices of IDs are "A" and "B"; "**Summary-A**" and "**Summary-B**" are replaced with model-generated texts. Since LLMs are not robust to ID and order (Zheng et al., 2023; Shen et al., 2023), we enumerate different combinations for a given pair, resulting in four LLM queries.

Please evaluate the overall quality of the following translations from English to Dutch.

Evaluation Criteria:
Overall Quality: A good translation should: 1) faithfully reflect the meaning of the source text; 2) avoid adding unnecessary or irrelevant details. 3) use natural and fluent Dutch.

Source: **[Source]**
Translation **[ID1]**: **[Translation-A]**
Translation **[ID2]**: **[Translation-B]**

FIRST, provide a one-sentence comparison of the two translations for overall quality, explaining which you prefer and why.
SECOND, on a new line, state only the ID to indicate your choice. Your response should use the format:
Overall Quality: ¡one-sentence comparison and explanation¿
Preferred: ¡translation ID¿

---

Please evaluate the informativeness of the following translations from English to Dutch.

Evaluation Criteria:
Informativeness: Does the translation preserve all key information without adding irrelevant details?

Source: **[Source]**
Translation **[ID1]**: **[Translation-A]**
Translation **[ID2]**: **[Translation-B]**

FIRST, provide a one-sentence comparison of the two translations for informativeness, explaining which you prefer and why.
SECOND, on a new line, state only the ID to indicate your choice. Your response should use the format:
Informativeness: ¡one-sentence comparison and explanation¿
Preferred: ¡translation ID¿

---

Please evaluate the coherence of the following translations from English to Dutch.

Evaluation Criteria:
Coherence: Is the translation fluent, logically structured, and easy to understand in Dutch?

Source: **[Source]**
Translation **[ID1]**: **[Translation-A]**
Translation **[ID2]**: **[Translation-B]**

FIRST, provide a one-sentence comparison of the two translations for coherence, explaining which you prefer and why.
SECOND, on a new line, state only the ID to indicate your choice. Your response should use the format:
Informativeness: ¡one-sentence comparison and explanation¿
Preferred: ¡translation ID¿

Table 10: Prompt templates for LLM evaluation on the machine translation task in terms of overall quality, informativeness, and coherence. Here, "**Source**" is the source sentence to be translated. The choices of IDs are "A" and "B"; "**Translation-A**" and "**Translation-B**" are replaced with model-generated texts. We still enumerate different combinations for a given pair, resulting in four LLM queries.

