# OpenReview forum: "BRIM: Block-wise Return Induction Method for Sequence Knowledge Distillation"
_ICLR.cc/2026/Conference — ICLR 2026 Conference Withdrawn Submission_

### Official Review · Reviewer_kvMT · 2025-10-27

**Soundness:** 3
**Presentation:** 2
**Contribution:** 2
**Rating:** 4
**Confidence:** 4

**Summary:**

The paper targets the high-variance problem in RL-based sequence KD. It proposes BRIM, which applies an inverse Bellman optimality expansion over K-step blocks along student rollouts to construct an approximate return used in policy-gradient updates. The paper positions BRIM as a REINFORCE-with-baseline variant whose baseline arises from teacher Q-values, with a variance-reduction argument. In the experiment section, it shows empirical gains across datasets including benchmarks of summarization, translation, and math reasoning.

**Strengths:**

* The paper proposes a novel approach to reduce variance induced in teacher-student distillation, with clear intuition.
* The method shows consistent empirical gains on T5 models compared to previous baselines across evaluation benchmarks.

**Weaknesses:**

* The presentation lacks clarity and could make the algorithm hard to follow. For example, the paper does not mention how Q value function is implemented in the teacher model. Please consider adding more clarifications about this part.
* All experiments use T5 for both teacher and student, which limits conclusions about model-family generality. Results on decoder-only families (e.g., Llama, Qwen, Mistral) and cross-family teacher-to-student settings are needed, along with a sensitivity study to weaker/miscalibrated teachers.
* The current evaluation tasks (seq2seq tasks + GSM8K) do not include open-ended instruction following or multi-turn settings, where sequence-level RL variance is often most problematic. Evaluations on mainstream instruction-following benchmarks(e.g., MT-Bench [1], AlpacaEval 2.0 [2]) would better validate the method’s breadth.
* The method is motivated as a low-variance alternative to REINFORCE-style sequence RL, yet there are no results for (i) REINFORCE on task reward and (ii) PPO+GAE on task reward under identical budgets. Adding REINFORCE based RL and PPO as baseline could better validate the method's effectiveness

[1] Zheng, Lianmin, Wei-Lin Chiang, Ying Sheng, Siyuan Zhuang, Zhanghao Wu, Yonghao Zhuang, Zi Lin et al. "Judging llm-as-a-judge with mt-bench and chatbot arena." Advances in neural information processing systems 36 (2023): 46595-46623.

[2] Dubois, Yann, Balázs Galambosi, Percy Liang, and Tatsunori B. Hashimoto. "Length-controlled alpacaeval: A simple way to debias automatic evaluators." arXiv preprint arXiv:2404.04475 (2024).

**Questions:**

* In line 154-160, How's $q(s, a)$ defined for teacher model ? Do you train any separate critic or value function, please clarify for the implementation for the $Q$ value function of teacher model.
* The current K-step return estimate might be closely related to the classical multi-step/bootstrapped estimators such as n-step return [1], TD($\lambda$) [2], could you elaborate on the difference between your method and existing methods to reduce the variance?

[1] Mnih, V., et al. (2016). Asynchronous Methods for Deep Reinforcement Learning (A3C). ICML 2016.

[2] Sutton, R. S. (1988). Learning to predict by the methods of temporal differences. Machine Learning 3(1): 9–44.

---

> ### Author Response · Authors · 2025-11-15
>
> We thank the reviewers for pointing out the novelty of our approach.
>
> >weakness 1: The presentation lacks clarity and could make the algorithm hard to follow. For example, the paper does not mention how Q value function is implemented in the teacher model. Please consider adding more clarifications about this part.
>
> We thank the reviewer for this question.
>
> To clarify, we do not train a separate Q-value function for the teacher model. **As stated in Line 95**, "a language model’s pre-softmax logit can be viewed as the Q-value function". In other words, we directly take the pre-activation values of the teacher’s prediction layer as the Q-values. This follows the standard practice in previous work [3,4].
>
> In our paper, we **provided a concise derivation for why an LLM’s logits can be used as Q-values (Lines 89--115)**, and the reviewer may refer to prior work [3,4] for details.
>
>
> > Weakness 2: All experiments use T5 for both teacher and student, which limits conclusions about model-family generality. Results on decoder-only families (e.g., Llama, Qwen, Mistral) and cross-family teacher-to-student settings are needed, along with a sensitivity study to weaker/miscalibrated teachers.
>
> We thank the reviewer for this question.
>
> In fact, the original manuscript presented additional experiments on other models (namely, Qwen) in Appendix F.  The content arrangement (main paper vs appendix) is due to space constraints.
>
> Overall, both T5 and Qwen show that our approach achieves consistent improvements across all datasets, which confirms the robustness and generalizability of the proposed BRIM approach.
>
> Regarding cross-family distillation: Cross-family distillation faces significant challenges due to different tokenizations. Although mapping different tokens itself is an important research direction (which we’re happy to explore as future work), this goes beyond the scope of our work. We’ve discussed this in the limitation paragraph in the revision.
>
> Weaker teacher: Thanks for the suggestion. We’re currently running an experiment, where the teacher is FLAN-T5-XL/large/base, and the student is FLAN-T5-small. We’re still at the stage of finetuning the teachers. We’ll report back when we have the results.
>
> >weakness 3: The current evaluation tasks (seq2seq tasks + GSM8K) do not include open-ended instruction following or multi-turn settings, where sequence-level RL variance is often most problematic. Evaluations on mainstream instruction-following benchmarks(e.g., MT-Bench [1], AlpacaEval 2.0 [2]) would better validate the method’s breadth.
>
>
> We thank the reviewer for this suggestion.
>
> We **conduct the KD experiments on the suggested benchmarks**. We use the same LLM-as-judge evaluation pipeline described in Section 3.3 to report the win rate against student distilled with the KL approach. The results are summarized in the following table:
>
> | Model | MT-Bench | AlpacaEval |
> | --- | --- | --- |
> | Teacher Model | 96.9 | 97.3 |
> | Student-TVD | 58.4 | 60.5 |
> | Student-LLMR | 62.1 | 57.1 |
> | Student-BRIM (ours) | 70.7 | 69.3 |
>
> These results show that our approach brings decent improvement compared with the competing approaches, which is also consistent with other results in the manuscript.
>
> >weakness 4: The method is motivated as a low-variance alternative to REINFORCE-style sequence RL, yet there are no results for (i) REINFORCE on task reward and (ii) PPO+GAE on task reward under identical budgets. Adding REINFORCE-based RL and PPO as baseline could better validate the method's effectiveness
>
> We thank the reviewer for raising this.
>
> (i) Results for REINFORCE: It’s noted that the competing approach **LLMR is indeed a REINFORCE-based RL**. Moreover, we experimented with other REINFORCE variants, including LLMR + Mean Baseline and LLMR + Min-Variance Baseline. The nature of these methods was described in Section 3.1.
>
> (ii) Results for PPO+GAE: We did conduct these experiments. We **have discussed the relationship between BRIM and PPO in Lines 452-467 and provide a detailed experimental analysis in Appendix D**. Results in Figure 3 show that our BRIM significantly outperforms PPO+GAE (which suffers from the collapse issue).
>
> >Question 1: In line 154-160, How's  defined for teacher model ? Do you train any separate critic or value function, please clarify for the implementation for the  value function of teacher model.
>
> Thank you for asking for this clarification. Please see our response to Weakness 1.

---

> > ### Author Response · Authors · 2025-11-15
> >
> > >Question 2: The current K-step return estimate might be closely related to the classical multi-step/bootstrapped estimators such as n-step return [1], TD() [2], could you elaborate on the difference between your method and existing methods to reduce the variance?
> >
> > Thank you for this insightful question.
> >
> > This is an important difference, which we have addressed in our related work discussion in Section 4. Here, we provide a comparison in point form.
> >
> > Generally, the difference between K-step return estimation and TD can be explained from both a mathematical perspective and a procedural perspective.
> >
> > 1. Different mathematical meanings of "N" in N-step bootstrapping and K in our approach.
> >     * The *N-step* bootstrapping is to use N-step return to bootstrap the current step's state-value function; in their approach, N and variance are **positively** correlated in a **linear** fashion.
> >     * *K* in our approach is calculated the total return based on K-step intervals; in our approach, K and variance are **negatively** correlated in a **power-law** fashion.
> > 2. No need for critic (the-state-value function).
> >     * N-step bootstrapping estimates the expected return by combining n-step look-ahead information **with** an additional critic.
> >     * Our work estimates the expected return by our proposed K-step return formulation in a sequential distillation scenario, **without** learning a critic function, which will cause training stability issues [5,6] (also discussed in Appendix D on Pages 18 &19).
> >
> > Overall, our K-step is a novel approach different from traditional TD (and other RL methods).
> >
> > > Summary
> >
> > We are grateful to the reviewer’s suggestions and are conducting additional experiments on weak teachers.
> >
> > Other than this, it appears that most of the questions have already been addressed in the appendix.
> >
> > We sincerely hope that the reviewer could revisit our paper (especially the appendices) and reassess our paper. Thank you very much!
> >
> >
> >
> > [3] Yongchang Hao, et al. 2023. Teacher forcing recovers reward functions for text generation. NIPS.
> >
> > [4] Dongheng Li, et al. 2024. LLMR: Knowledge distillation with a large language model-induced reward. COLLING.
> >
> > [5] Shalabh Bhatnagar, et all.  2007. Incremental natural actor-critic algorithms. NIPS.
> >
> > [6] Simone Parisi, et al. 2019. TD-Regularized Actor-Critic methods. Machine Learning.

---

> ### Comment · Reviewer_kvMT · 2025-11-28
>
> I thank the authors for their clarifications and responses. I have the following follow-ups:
> * For $q(s,a)$ used for student model training, please consider clarifying in the algorithm section (Section 2.2) exactly how it is computed from the teacher; For Qwen(Qwen1.5) results currently in the appendix, please include them in the main text alongside the T5 experiments, and consider including more recent families (e.g., Qwen2.5, Qwen3) for completeness.
> * Your reply notes that LLMR is REINFORCE-based, but it remains a teacher-reward baseline. The key question in my original review is about REINFORCE or PPO+GAE on the task reward (teacher-free), to assess the method's robustness.
> * I appreciate the explanation comparing the $K$-step block return with $n$-step/TD methods. Conceptually, the stride-$K$ block return construction could be closely related to temporal abstraction, a variant of $n$-step bootstrapping applied on a coarsened timeline, which is well studied in the options/SMDP literature[1]. Further clarifying the $K$-step block return’s connection to the prior work could help strengthen the novelty claim.
>
> [1] Sutton, Richard S., Doina Precup, and Satinder Singh. "Between MDPs and semi-MDPs: A framework for temporal abstraction in reinforcement learning." Artificial intelligence 112, no. 1-2 (1999): 181-211.

---

### Official Review · Reviewer_Qefc · 2025-10-28

**Soundness:** 2
**Presentation:** 3
**Contribution:** 2
**Rating:** 4
**Confidence:** 4

**Summary:**

This paper proposes a novel method called BRIM (Block-wise Return Induction Method) to address the high-variance issue in Reinforcement Learning (RL)-based Knowledge Distillation (KD) for text generation, which is caused by long action sequences during sampling. The method segments the student model's generated trajectory into blocks of length K. By applying the inverse Bellman Optimality Equation to each block, it induces a block-wise cumulative reward from the teacher model, which serves as the training signal for policy gradient. Theoretical analysis demonstrates that this approach effectively reduces the variance of gradient estimates. Extensive experiments on three text generation tasks (summarization, machine translation, and arithmetic reasoning) validate its superior performance in both standard task metrics and LLM-based evaluation.

**Strengths:**

S1: BRIM applies the inverse Bellman equation for block-wise reward induction, effectively mitigating the high variance problem in long-sequence RL training, which is novel and well-grounded in theory.
S2: The paper proposes the method with a theoretical proof that it reduces gradient variance (Theorem 1) and analyzes the bias-variance trade-off, which significantly enhances the credibility of the approach.
S3: The method is systematically evaluated on three text generation tasks from different domains. The evaluation includes not only traditional metrics (e.g., ROUGE, BLEU) but also introduces an LLM-based assessment, verifying the method's generality and effectiveness.
S4: The varianc-bias trade-off trends observed experimentally match theoretical predictions.

**Weaknesses:**

W1: It seems that the main content of this work is a simplified estimation of the calculation of G_t in [1], so I hope to see more insights from the author about this work and further explanation of possible optimizations.
W2: Theorem 1 relies on the assumption that the (state, action, reward) tuples are independent and identically distributed across timesteps. However, in autoregressive text generation, such tuples are inherently correlated since each token depends on its preceding context. The paper only briefly mentions this issue; therefore, a more thorough discussion is needed to justify when this i.i.d. assumption can be approximately valid. For example, when the student policy closely matches the teacher’s distribution or when large-batch sampling mitigates correlation effects.
W3: The experimental results show that the optimal value of K is inconsistent across different tasks and datasets (e.g., 2, 4, 8, 16). I hope the author could provide a strategy for automatically selecting K, which could increase tuning costs in practical applications.
W4: Estimation bias in the derivation process of Eq 5: In addition to the bias pointed in line 134, the decoding sampling strategy also has an impact, unless greedy decoding is used. Moreover, as K increases, the bias accumulates further. What kind of impact does this produce? From Fig. 1, there does not seem to be a consistent effect.


[1] LLMR: Knowledge Distillation with a Large Language Model-Induced Reward

**Questions:**

Please refer to weakness.

---

> ### Author Response · Authors · 2025-11-15
>
> We thank the reviewer for pointing out a number of strengths, including the novelty and theoretical value of our work.
>
> > Weakness 1: It seems that the main content of this work is a simplified estimation of the calculation of G_t in [1], so I hope to see more insights from the author about this work and further explanation of possible optimizations.
>
> Thank you for the questions.
>
> Our insights are that the traditional method [1] estimates G_t by the classic REINFORCE algorithm, which suffers from a **high variance issue**. Our method provides a novel estimate of G_t that **largely reduces the variance** and thus improves RL training.
>
> In particular, our **unique insight** lies in the observation that, with block-wise Bellman optimality, intermediate terms are cancelled shown in Eqn. (5,6,7), so the variance is reduced (proved by Theorem 1 and empirically verified by experiments in Table 2). In the revision, we’ve elaborated such insight before Theorem 1.
>
> For “possible optimizations”, we would like to seek clarification on what "optimization" means by the reviewer:
>
> * If the reviewer is asking about parameter optimization, we train the model by policy gradient and **the detailed pseudocode is provided in Algorithm 1 (Lines 972-995)**.
>
> * If the reviewer is asking about further optimizing G_t estimation, we’re indeed exploring adaptive K-step estimation, where K is chosen dynamically based on exploration/exploitation trade-off. Since this becomes a different method, we’ve mentioned it as future work in the revision. Thanks for the suggestion!
>
>
>
> >Weakness 2: Theorem 1 relies on the assumption that the (state, action, reward) tuples are independent and identically distributed across timesteps. However, in autoregressive text generation, such tuples are inherently correlated since each token depends on its preceding context. The paper only briefly mentions this issue; therefore, a more thorough discussion is needed to justify when this i.i.d. assumption can be approximately valid. For example, when the student policy closely matches the teacher’s distribution or when large-batch sampling mitigates correlation effects.
>
> We appreciate the reviewer's observation.
>
> Admittedly, our theorem is based on the iid assumption. This is, in fact, widely adopted in theoretical RL analysis [2,3,4]. It is a reasonable assumption in practice, because the dependencies between distant tokens decay rapidly, and this correlation is further weakened when a large batch of samples is used for training.
>
> Crucially, our empirical results in Table 2 show a clear reduction in variance, which aligns perfectly with the theoretical predictions of Theorem 1. This suggests our assumption is reasonable.
>
> We also thank the reviewer for suggesting a more thorough discussion, for example, (1) “when the student policy closely matches the teacher’s distribution”, and (2) “when large-batch sampling mitigates correlation effects.”
>
> For point (1), our original paper already provided a theoretical analysis showing that the resemblance between the teacher and student reduces the bias (Theorem 1 in Appendix B). We encourage the reviewer to double-check our paper (including the appendix) and let us know if you have any further questions. Thanks!
>
> For point (2), we totally agree with the reviewer and have discussed this after Theorem 1.
>
>
> >Weakness 3: The experimental results show that the optimal value of K is inconsistent across different tasks and datasets (e.g., 2, 4, 8, 16). I hope the author could provide a strategy for automatically selecting K, which could increase tuning costs in practical applications.
>
> This is a very practical question.
>
> As we have explored in Appendix G, a single optimal K may not be feasible because the approximation error depends on both block size K and sequence length T. Nevertheless, We found a clear correlation between the task's maximum output length and the optimal K (presented in Appendix G):
>
> | Models         | Xsum (max_output_len=64) | Europal (max_output_len=64) | GSM8K (max_output_len=256) |
> |----------------|--------------------------|-----------------------------|-----------------------------|
> | **T5 optimal K**   | 2                        | 2                           | 8                           |
> | **Qwen optimal K** | 2                        | 2                           | 16                          |
>
> Our finding can guide the choice of K in practice. With this guidance, we performed validation to obtain the optimal K in our experiments.

---

> > ### Author Response · Authors · 2025-11-15
> >
> > >Weakness 4: Estimation bias in the derivation process of Eq 5: In addition to the bias pointed in line 134, the decoding sampling strategy also has an impact, unless greedy decoding is used. Moreover, as K increases, the bias accumulates further. What kind of impact does this produce? From Fig. 1, there does not seem to be a consistent effect.
> >
> > Thank you for your concern. We'd like to clarify these points:
> >
> > “Unless greedy decoding is used”: Indeed, our experiments used greedy decoding, as mentioned in Line 253. This can also be verified by the provided anonymous code. Thus, this concern is fully addressed.
> >
> > “What kind of impact does (bias from K) this produce?”: In fact, K controls the bias-variance trade-off of BRIM: a larger K typically indicates less variance but more bias. Empirically, a moderate value of K (e.g., 2--8) reduces variance to a significant extent without introducing too much bias, thus eventually yielding high RL performance.
> >
> > Fig. 1 “does not seem to be a consistent effect”: The results in Figure 1 are generally consistent, as the average return first increases and then decreases across all three tasks, when K increases from 1 (i.e., LLMR) to 16. However, the peak may not align exactly due to the output length and task specificity, which is understandable.
> >
> > To better demonstrate the consistent trend, we modified the x-axis labels in Figure 1 in the revision.
> >
> > > Summary
> >
> > We thank the reviewer again for the detailed review and for recognizing multiple key strengths of our work.
> >
> > In the author response, we have carefully responded to each of the points in detail and revised our manuscript accordingly. We hope our clarifications dispel any remaining misunderstandings. We would be grateful if you could take this additional context into account when revisiting your assessment. Thank you!
> >
> > [2] Michael J Kearns, et al. 2000. Bias-variance error bounds for temporal difference updates. CCLT.
> >
> > [3] Jalaj Bhandari, Daniel Russo, et al. 2018. A finite time analysis of temporal difference learning with linear function approximation. CCLT.
> >
> > [4] Tengyu Xu,et al. 2020. Reanalysis of variance reduced temporal difference learning. ICLR.

---

### Official Review · Reviewer_eUcQ · 2025-10-31

**Soundness:** 3
**Presentation:** 3
**Contribution:** 3
**Rating:** 6
**Confidence:** 3

**Summary:**

This paper proposes an improved RL approach called BRIM, which mitigates the high variance issue in knowledge distillation (KD). Specifically, the authors define the sum of rewards over consecutive steps by approximating that an optimal action is taken by a student policy. Based on this, they propose a K-step reward formulation for RL-based generation KD and update the model following the policy gradient formula. Experiments are conducted on various tasks, including XSum Summarization, Europarl EN-NL Translation, and GSM8K. The results demonstrate that BRIM with a larger K leads to a more stable training process and achieves better performance across these tasks.

**Strengths:**

* The paper is well-organized and clearly presented.
* The proposed approach is simple yet effective in mitigating the variance issue in RL-based knowledge distillation, and is supported by theoretical analysis.
* BRIM demonstrates consistent improvements over the baseline methods across different tasks presented in Table 1.

**Weaknesses:**

* The evaluation metrics in Table 1 are primarily based on n-gram matching. Incorporating semantic-centric metrics, such as G-Eval, could further validate the effectiveness of the proposed approach.
* The study lacks human evaluation. Although LLM-as-a-judge was used as a surrogate, LLM judges are prone to various biases and may not accurately reflect genuine human preferences.
* The experiments were conducted exclusively on T5 models with fewer than 3B parameters. Extending the evaluation to other model families and a wider range of teacher-student sizes is necessary to assess the robustness and generalizability of BRIM.

**Questions:**

1. It would be valuable to include an analysis of BRIM in knowledge distillation scenarios where there is a growing capability gap between the teacher and student models.
2. It is recommended to add experiments with other backbone language models, along with an analysis of their scaling trends.
3. Human evaluation should be incorporated to complement the automated metrics.
4. The addition of semantic-based evaluation metrics and the 95% confidence intervals for the results in Table 1 is suggested. While the authors state that their results are statistically significant compared to each baseline, the improvements on the Europarl dataset appear quite marginal, and some values seem to be bolded incorrectly.

---

> ### Author Response · Authors · 2025-11-15
>
> We thank the reviewer for recognizing the value of our work and also saying “the paper is well-organized and clearly presented.”
>
> > Weakness 1: ... Incorporating semantic-centric metrics, such as G-Eval, could further validate the effectiveness of the proposed approach.
>
> We appreciate this valuable suggestion.
>
> We would like to kindly point the reviewer to **Table 3 (Lines 410--416) in our paper, which already contains semantic-centric evaluation**. In particular, the reviewer suggests G-Eval, which is an LLM [evaluation platform](https://deepeval.com/docs/metrics-llm-evals). Our LLM-as-a-judge evaluation (although not using the G-Eval platform) has already provided the suggested semantic-centric evaluation.  The results in Table 3 show that BRIM significantly outperforms competing approaches on this semantic-oriented evaluation.
>
> >Weakness 2: The study lacks human evaluation ... LLM judges are prone to various biases and may not accurately reflect genuine human preferences.
>
> Thanks for the suggestion.
>
> We acknowledge that human evaluation can be another layer of evaluation, but it is costly (e.g., multiple annotators read hundreds of samples) and requires ethical approvals. Still, human evaluation may be very noisy and biased [1,2].  On the contrary, we use LLM-as-a-judge as a surrogate, but have evaluated the entire dataset. The results are consistent with string matching metrics and the RL metrics, showing that our evaluations are stable and reliable.
>
> Moreover, we **have already been mindful of potential LLM biases, as explained in Lines 406-408**. We took specific measures to address this issue, e.g., enumerating all combinations of answer IDs and order to prevent position and ID bias. Therefore, the bias of LLM-as-a-judge is  much mitigated.
>
> That being said, we are grateful to the reviewer’s suggestion and have included the discussion as a limitation paragraph.
>
>
> >Weakness 3: The experiments were conducted exclusively on T5 models with fewer than 3B parameters. Extending the evaluation to other model families and a wider range of teacher-student sizes is necessary to assess the robustness and generalizability of BRIM.
>
>
> In fact, **the original manuscript presented additional experiments on other models (namely, Qwen) in Appendix F**, where the teacher model is larger than 3B.  The content arrangement (main paper vs appendix) is due to space constraints.
>
> Overall, both T5 and Qwen show that our approach achieves consistent improvements across all datasets, which confirms the robustness and generalizability of the proposed BRIM approach.
>
> >Question1: ... include an analysis of BRIM in knowledge distillation scenarios where there is a growing capability gap between the teacher and student models.
>
> Thank you for this insightful question.
>
> We chose the student and teacher model sizes by following [4,5,6] for fair comparison.
>
> We’re now conducting a model size analysis as suggested by the reviewer (thanks!), and will report back once we have the results.
>
> >Question 2: ... add experiments with other backbone language models
>
> Thanks. The original manuscript already reported Qwen results. See details in response to Weakness 3.
>
>
> > Question 3: Human evaluation should be incorporated to complement the automated metrics.
>
> See our response to Weakness 2. Thank you!
>
> > Question 4: The addition of semantic-based evaluation metrics and the 95% confidence intervals for the results in Table 1 is suggested. While the authors state that their results are statistically significant compared to each baseline, the improvements on the Europarl dataset appear quite marginal, and some values seem to be bolded incorrectly.
>
> Thank you for the questions.
>
> Semantic-based evaluation metrics: See our response to Weakness 1. Thanks!
>
> Confidence Intervals: **Our original manuscript already included 95% confidence intervals for all metrics in Table 1 **(including ROUGE, BLEU, chrF, and TER), as suggested by the reviewer. This has already been **mentioned in our experimental setup description (Line 269)**.
>
> Bolding incorrect numbers: We double checked our tables and did not find errors. We’ll be grateful if the reviewer could point out which number is wrong.
>
> > summary
>
> We sincerely thank the reviewer for recognizing the scientific value of our work. We have provided a thorough response to every concern raised by the reviewer. It’ll be highly appreciated if the reviewer could increase the score if our responses are helpful for clarifying any misunderstanding. Please feel free to let us know if you have any further questions.

---

> > ### Author Response · Authors · 2025-11-15
> >
> > [1] Hosking, T. et al. 2024 Human Feedback is not Gold Standard. ICLR.
> >
> > [2] Thomson, C. & Reiter, E, 2024 Common Flaws in Running Human Evaluation in NLP. CL.
> >
> > [3] Yongchang Hao, et al. 2022. Teacher forcing recovers reward functions for text generation. NIPS.
> >
> > [4] Yuqiao Wen, et al. 2023. f-divergence minimization for sequence-level knowledge distillation. ACL.
> >
> > [5] Dongheng Li, et al. 2024. LLMR: Knowledge distillation with a large language model-induced reward. COLLING.
> >
> > [6] Rishabh Agarwal, et al. 2024. On-policy distillation of language models: Learning from self-generated mistakes. ICLR.

---

> > ### Author Response · Authors · 2025-11-24
> >
> > We thank the reviewer for this insightful suggestion.
> >
> > We conducted additional experiments varying the student model size to explicitly manipulate the capability gap. Using the same Teacher model as in Section 3.2, we distilled knowledge into three distinct student sizes: 77M (Large Gap), 250M (Medium Gap), and 800M (Small Gap).
> >
> > The results demonstrate that BRIM remains effective even when the capability gap is significant.
> >
> > | Model Size     | XSum  | Europarl | GSM8K | Observations                                      |
> > |----------------|-------|----------|-------|---------------------------------------------------|
> > | Student-77M    | 32.04 | 22.05    | 14.86 | Retains stability despite extreme compression     |
> > | Student-250M   | 36.63 | 22.95    | 26.38 | Consistent scaling behavior                       |
> > | Student-800M   | 39.82 | 23.23    | 33.43 | Approaches Teacher performance                    |

---

> > > ### Comment · Reviewer_eUcQ · 2025-11-27
> > > **Follow-up Questions**
> > >
> > > Thank the authors for their detailed response. My concerns are as follows:
> > >
> > > 1. I could not find the 95% confidence intervals for all metrics in Table 1. Could you please check the manuscript or provide further explanation?
> > >
> > > 2. How about evaluating the performance on more recent models, such as Qwen3?
> > >
> > > 3. Human evaluation is still expected. Using LLM-as-a-judge may introduce unintended biases and can be easily misled [1].
> > >
> > > [1] Zheng, Xiaosen, et al. "Cheating automatic llm benchmarks: Null models achieve high win rates." arXiv preprint arXiv:2410.07137 (2024).

---

### Official Review · Reviewer_ZqJx · 2025-11-01

**Soundness:** 1
**Presentation:** 2
**Contribution:** 2
**Rating:** 4
**Confidence:** 3

**Summary:**

This paper proposes a Block-wise Return Induction Method (BRIM) for reinforcement learning (RL)-based knowledge distillation. BRIM mainly based on LLMR (Li et al. 2024). BRIM further introduces a K-step reward estimation. This paper evaluates BRIM on three benchmarks including XSum (for summarization tasks), Europarl (for translation tasks), and GSM8K (for mathematical reasoning tasks).

**Strengths:**

- S1. [Idea] This paper aims to advance RL-based knowledge distillation methods, and these approaches seem promising in training small language models.

**Weaknesses:**

- W1. [Novelty] BRIM mainly based on LLMR (Li et al. 2024). LLMR is a knowledge distillation method based on a reward function induced from large language models. Based on LLMR, BRIM further introduces a K-step reward estimation. However, the extension seems rather limited. Furthermore, it is unclear what the advantages of K-step reward estimation are.

- W2. [Performance] According to Table 1, BRIM (26.38 on GSM8K) does not seem to have a significant difference in performance from LLMR (25.39 on GSM8K).

- W3. [Evaluation] This paper evaluates BRIM on three benchmarks including XSum, Europarl, and GSM8K. These are rather easy benchmarks in each task. I am not sure that BRIM works well on more complex benchmarks such as AIME2024 instead of GSM8K.

**Questions:**

- Q1. What are the advantages of K-step reward estimation of BRIM, compared to LLMR?

---

> ### Author Response · Authors · 2025-11-15
>
> We thank the reviewer for saying that “[our] approaches seem promising”.
>
> > weakness1 [Novelty] BRIM mainly based on LLMR (Li et al. 2024). ... Based on LLMR, BRIM further introduces a K-step reward estimation. However, the extension seems rather limited. Furthermore, it is unclear what the advantages of K-step reward estimation are.
>
> Thank you for the questions.
>
> Regarding the novelty: we kindly refer the reviewer to Lines 47--67, where we detail the novelty of our approach. Our BRIM significantly differs from LLMR, as LLMR is a direct application of the classic REINFORCE algorithm. On the other hand, our BRIM proposes a **novel RL algorithm**: it falls into the REINFORCE-with-baseline category, but derives a **novel baseline term** by the block-wise Bellman Optimality Equation (Eq. 11). We further provide **theoretical analysis** on the variance and bias of our algorithm (Theorem 1). Therefore, our contribution is not limited.
>
> Regarding the advantages: The primary advantage of our BRIM, as summarized in the abstract (Line 15), is that it "mitigates the high variance issue and stabilizes the training process". This claim is (1)
> **theoretically justified** as we provide a formal analysis in Theorem 1 (Variance Reduction), and (2) **empirically verified** as our in-depth analyses (Lines 355-400) confirm this, with Table 2 demonstrating the variance mitigation and Figure 2 illustrating the resulting training stability.
>
> Consequently, our BRIM improves the task performance (Figure 1 and Table 1).
>
>
> >weakness2 Performance] According to Table 1, BRIM (26.38 on GSM8K) does not seem to have a significant difference in performance from LLMR (25.39 on GSM8K).
>
> Thanks for raising the point.
>
> We agree that the ~1% absolute improvement on GSM8K (26.38 vs. 25.39) appears modest at first glance. However, we’d like to place this result in the context of our experimental setup:
>
> Model Scale: This 1% gain is achieved by a 250MB student model distilling from a 3B teacher. With this model scale, LLMR [3] beats TVD [2] by only 0.3% (25.21% vs 24.94%). Under this comparison, **1% gain is a decent improvement**.
>
> Relative Performance: As noted in [1], the same 250MB student model achieves 27.2% accuracy only when trained with CoT data from a much more powerful 175B (GPT-3.5) teacher. Our BRIM only uses a 3B teacher but achieves an accuracy of 26.38%, showing that the student performance on GSM8K is nearly saturated. Nevertheless, we achieve 1% improvement, which should be viewed as a **considerable gain**.
>
> Consistency: Most importantly, BRIM's performance improvement is consistent across three distinct tasks (Summarization, Machine Translation, and Math Reasoning) on both task-oriented metrics (Table 1) and the RL metric (Figure 1).
>
> Overall, we believe this 1% absolute improvement on GSM8K is meaningful and decent, and that our BRIM is an effective approach in general.
>
> >weakness3 This paper evaluates BRIM on three benchmarks, including XSum, Europarl, and GSM8K. These are rather easy benchmarks in each task. I am not sure that BRIM works well on more complex benchmarks such as AIME2024 instead of GSM8K.
>
> We’re grateful to the reviewer’s constructive suggestion.
>
> We chose XSum, Europarl, and GSM8K because this **ensures a fair and direct comparison with the competing approaches** [2,3,4] cited in our paper, as they also used these datasets.
>
> Following the reviewer's suggestion, we **conducted a new experiment using the AIME2024 benchmark**, a challenging math dataset consisting of only 30 datapoints. We first evaluated our 3B teacher model (FLAN-T5 XL) on this dataset. Despite achieving 40.71% on GSM8K, it gave  0% accuracy on the AIME2024 set. We then attempted to fine-tune the teacher on the AIME2024 with 20 training and 10 validation; however, even after 20 epochs, the validation accuracy remained 0%. Likewise, 30 datapoints do not seem to be enough for knowledge distillation.
>
> This new finding suggests that AIME2024 is a viable option for this paper, but we will keep it in mind in other studies. Thanks again!
>
> >question 1. What are the advantages of BRIM, compared to LLMR
>
> Thanks for the question. Please see our response to Weakness 1.
>
> > Summary
>
> We sincerely thank the reviewer for the effort. In our author response, we have provided a thorough response to every concern raised by the reviewer. We’ll be happy to answer any further questions and are looking forward to your support!
>
>
> [1] Tianduo Wang, et al. 2024 Self-training with direct preference optimization improves chain-of-thought reasoning. ACL.
>
> [2] Yuqiao Wen, et al. 2023. f-divergence minimization for sequence-level knowledge distillation. ACL.
>
> [3] Dongheng Li, et al. 2024. LLMR: Knowledge distillation with a large language model-induced reward. COLLING.
>
> [4] Rishabh Agarwal, et al. 2024. On-policy distillation of language models: Learning from self-generated mistakes. ICLR.

---

### Author Response · Authors · 2025-11-24
**Reminder for reviewer-author discussion**

Dear Reviewers:

We thank your efforts in reviewing our paper. It has been a while since we provided an initial response. However, we have not seen follow-up comments yet.

In our author response, we've clarified all the points. We also provided additional experiments to address the concerns.

It'll be highly appreciated if you could take a look at our responses and let us know if there’re any additional concerns.  We’re happy to discuss further and revise our manuscript accordingly.


Thank you very much,
-Authors

---

### Note · Authors · 2025-12-11

I have read and agree with the venue's withdrawal policy on behalf of myself and my co-authors.